# Robust spin correlations at high magnetic fields in the harmonic honeycomb iridates

K.A. Modic[1,5], B.J. Ramshaw[1,6], J.B. Betts[1], Nicholas P. Breznay [2,3], James G. Analytis[2,3], Ross D. McDonald[1] & Arkady Shekhter [4]

The complex antiferromagnetic orders observed in the honeycomb iridates are a double-edged sword in the search for a quantum spin-liquid: both attesting that the magnetic interactions provide many of the necessary ingredients, while simultaneously impeding access. Focus has naturally been drawn to the unusual magnetic orders that hint at the underlying spin correlations. However, the study of any particular broken symmetry state generally provides little clue about the possibility of other nearby ground states. Here we use magnetic fields approaching 100 Tesla to reveal the extent of the spin correlations in $\gamma$-lithium iridate. We find that a small component of field along the magnetic easy-axis melts long-range order, revealing a bistable, strongly correlated spin state. Far from the usual destruction of antiferromagnetism via spin polarization, the high-field state possesses only a small fraction of the total iridium moment, without evidence for long-range order up to the highest attainable magnetic fields.

[1] Los Alamos National Laboratory, Los Alamos, NM 87545, USA. [2] Materials Science Division, Lawrence Berkeley National Laboratory, Berkeley, CA 94720, USA. [3] Department of Physics, University of California, Berkeley, CA 94720, USA. [4] National High Magnetic Field Laboratory, Florida State University, Tallahassee, FL 32310, USA. [5] Present address: Max-Planck-Institute for Chemical Physics of Solids, Noethnitzer Strasse 40, Dresden D-01187, Germany. [6] Present address: Laboratory for Atomic and Solid State Physics, Cornell University, Ithaca, NY 14853, USA. Correspondence and requests for materials should be addressed to K.A.M. (email: modic@cpfs.mpg.de)

Spin systems with highly anisotropic exchange interactions have recently been proposed to host quantum spin-liquid states[1]. It has been suggested that the extreme exchange anisotropy required to achieve such states can be mediated by the strong spin–orbit interactions of transition-metal ions situated in undistorted and edge-sharing oxygen octahedra[2]. Both the two-dimensional ($\alpha$) and three-dimensional (3D) ($\beta$ and $\gamma$) polymorphs of the honeycomb iridates ($A_2IrO_3$, A = Na, Li) closely fulfill these geometric requirements and display intriguing low-field magnetic properties that indirectly indicate the presence of anisotropic exchange interactions between the iridium spins[3–5]. Kitaev's spin-liquid ground state, however, results from the presence of exclusively bond-specific Ising exchange interactions[1], which have not yet been verified experimentally. To date, all honeycomb iridates deviate sufficiently from this ideal, such that they transition to long-range magnetic order at finite temperature[6,7] drawing attention to the complex magnetic structures of their ground states[8–13].

$\gamma$-lithium iridate features an incommensurate magnetic structure with non-coplanar and counter-rotating moments below $T_N = 38$ K[6]. The magnetic anisotropy within the ordered phase was extensively characterized in our previous study[4]. With increasing magnetic field below $T_N$, the magnetic torque $\tau$, divided by the applied field $H$ increases linearly up to an angle-dependent field $H^*$, defining the phase boundary of the low-field ordered state[4]. Importantly, the sharp feature at $H^*$ is not accompanied by full saturation of the iridium moment[4]. Instead, the magnetic moment at $H^*$ is only ~0.1 $\mu_B$[4]. The lack of a fully saturated moment at $H^*$ implies either the onset of another magnetically ordered phase above $H^*$, or alternatively a transition into a paramagnetic state lacking long-range order. The latter implies that the spin correlations are controlled by exchange interactions much stronger than the applied magnetic field. Just as the strange metallic state near the quantum critical point in the hole-doped cuprates is revealed once superconductivity is suppressed with magnetic field[14], we use high magnetic fields to destroy the antiferromagnetic order and expose the spin correlations in $\gamma$-lithium iridate.

In the following, we show that above $H^*$ $\gamma$-$Li_2IrO_3$ is characterized by a highly anisotropic nonlinear magnetic response with a small net magnetic moment up to the highest attainable magnetic fields. The absence of a sharp boundary with the high-temperature paramagnetic phase suggests that the high-field state does not break additional symmetries and does not exhibit long-range order. Furthermore, a hysteresis-free magnetic anisotropy that can be tied to the crystallographic directions evolves continuously with field out of the ordered state, uncharacteristic of glassy behavior. With these features in mind, and in light of recent studies that find dynamic evidence for liquid behavior in related transition-metal honeycomb structures[15,16], we will refer to the observed high-field state as a spin-fluid.

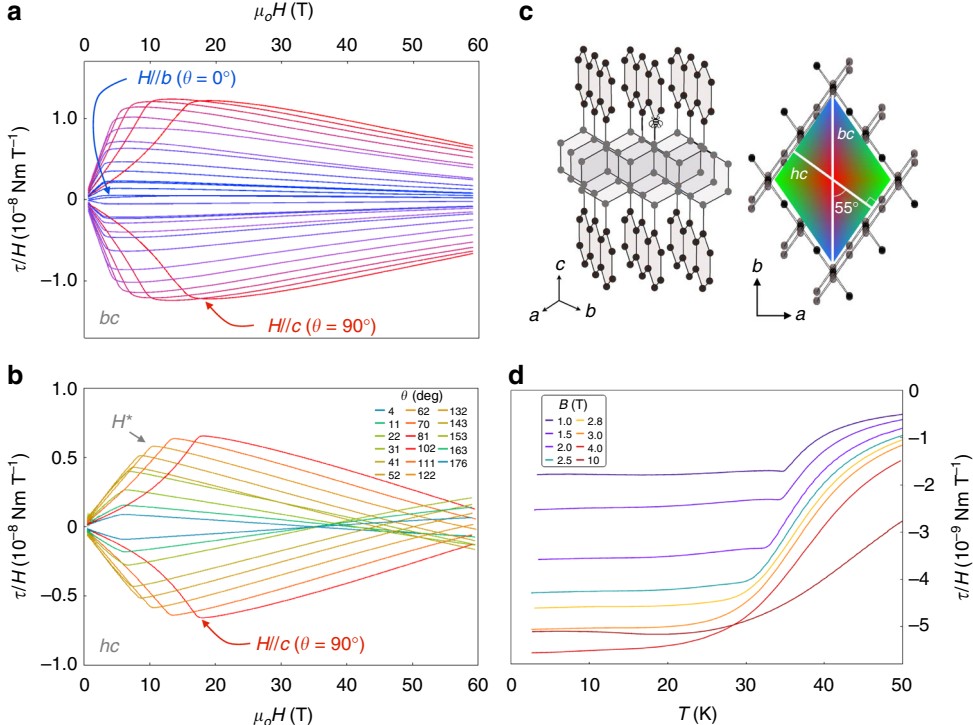

**Fig. 1** Crystal structure and anisotropic magnetic behavior of $\gamma$-lithium iridate. **a, b** Magnetic anisotropy, represented by $\tau/H$, as a function of magnetic field $H$ in the $bc$- and $hc$-planes at 4 K. A sharp and highly anisotropic feature defined as $H^*$ marks the suppression of long-range magnetic order and entry into a spin-fluid state. The angle $\theta$ in the legend is defined between the external magnetic field direction and the $ab$-plane such that $\theta = 90°$ corresponds to magnetic field aligned along the $c$-axis in both data sets. **c** The three-dimensional crystal structure of $\gamma$-lithium iridate, comprised of two orientations of iridium honeycomb planes. The diamond-shaped schematic of the crystal morphology is a reflection of the crystal structure when viewed along the $c$-axis. The crystal is rotated in **a, b** with respect to the external magnetic field in the planes depicted by the white lines in **c**, which also illustrate the orientation of the $bc$- and $hc$- rotational planes with respect to the two interwoven honeycomb planes. The colors on the diamond represent the direction of the applied magnetic field with respect to the crystallographic directions ($a$ = green, $b$ = blue and $c$ = red) for the measurements shown in **a, b**, with the color gradient mapping field direction for those angles between the principal directions. **d** $\tau/H$ as a function of temperature with magnetic field applied ~5° from the $b$-axis shows a crossover from a sharp transition to long-range order at fields below $H^*$ to a smooth change in torque that characterizes the onset of spin correlations at fields above $H^*$

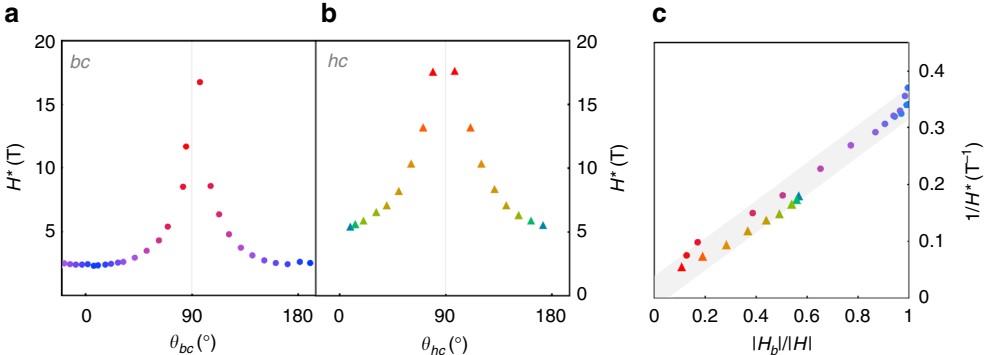

**Fig. 2** Angle dependence of $H^*$. **a** $H^*$ vs. rotation angle $\theta$ for both the $bc$- and **b** $hc$- rotational planes, with $\theta$ defined as the angle between the $ab$-plane and the magnetic field. **c** $1/H^*$ plotted vs. $|H_b|/|H|$. The magnitude of $1/H^*$ collapses approximately to a straight line for field rotation in both planes, illustrating that $H^*$ depends entirely on $M_b$

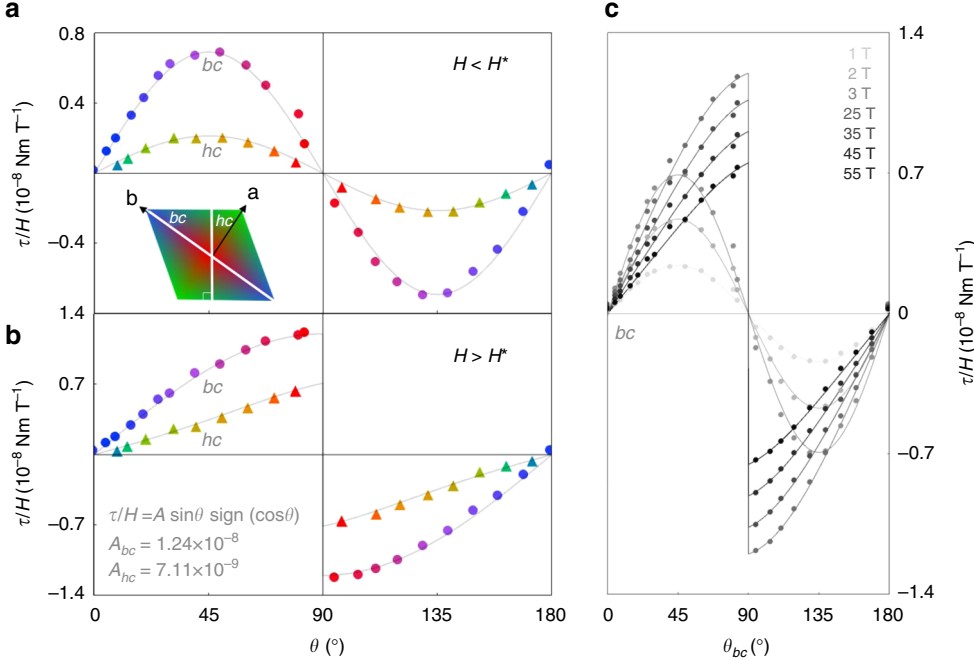

**Fig. 3** Evolution of the angle dependence of $\tau/H$ with increasing magnetic field at low temperatures. **a** At 4 K, $\tau/H$ evolves from a characteristic sin $2\theta$ angle dependence at small fields to **b** a correlation-driven sin $\theta$ sign(cos $\theta$) angle dependence at high fields. The discontinuity in $\tau/H$ across the $c$-axis ($\theta = 90°$) implies Ising-like behavior of the magnetization along the crystallographic $b$-direction. **c** The angle dependence of $\tau/H$ in the $bc$-plane at discrete field values shows a clear deviation from sin $2\theta$ behavior that persists up to 55 T

## Results

**Experimental technique.** We use torque magnetometry to study the high-field state of single crystal $\gamma$-lithium iridate. Specifically, we examine the nonlinear response at high magnetic fields. The anisotropy of the magnetic susceptibility $\alpha_{ij} = \chi_i - \chi_j$ in the linear response regime leads to a smooth sin $2\theta$ angle dependence with the torque vanishing when field is applied along the high symmetry directions (Supplementary Information of ref. [4]). Deviations from this smooth angle dependence provide a direct probe of magnetic correlations. The extreme magnetic anisotropy of $\gamma$-lithium iridate necessitated that both smaller volume samples and stiffer cantilevers were employed for the pulsed high-field measurements compared to our prior low-field (DC magnet) measurements[4]. To this end, we utilized focused ion beam lithography to cut and precisely align samples on Seiko PRC120 piezoresistive levers. This had the added benefit of reducing the lever deflection with field and systematic angle offsets, particularly when field is aligned close to the magnetically hard-axis.

**Magnetic anisotropy.** Figure 1a, b show $\tau/H$ for field rotation in two planes that include the $c$-axis: the $bc$-plane ($= 0°$) and a plane that is perpendicular to one of the honeycomb planes, referred to as the $hc$-plane ($= 55°$) (Fig. 1c). We find that $H^*$ closely follows a $1/|\cos\theta\cos\varphi|$ angle dependence (Fig. 2a, b), where $\theta$ denotes the angle between the $ab$-plane and the applied field in both rotation sets. The collapse of $1/H^*$ vs. the normalized $b$-component of magnetic field $|H_b|/|H| = |\cos\theta, \cos\varphi|$ onto a straight line in both rotation planes (Fig. 2c) indicates that the torque at $H^*$ is dominated by the $b$-component of magnetization $M_b$ and that the absolute value of magnetization $M(H^*)$ is nearly independent of field orientation. Moreover, the absolute value in the denominator of the angle dependence of $H^*$ indicates bistable behavior, $M_b = M_b^* \text{sign}(H_b)$. To determine whether the bistable behavior of $M_b$ persists to fields above $H^*$, we turn to the angle dependence of $\tau/H$ across the entire field range.

Below $H^*$, $\tau/H$ follows a sin $2\theta$ dependence (Fig. 3a) that is characteristic of the linear response regime, where $M_i = \chi_{ij}H_j$

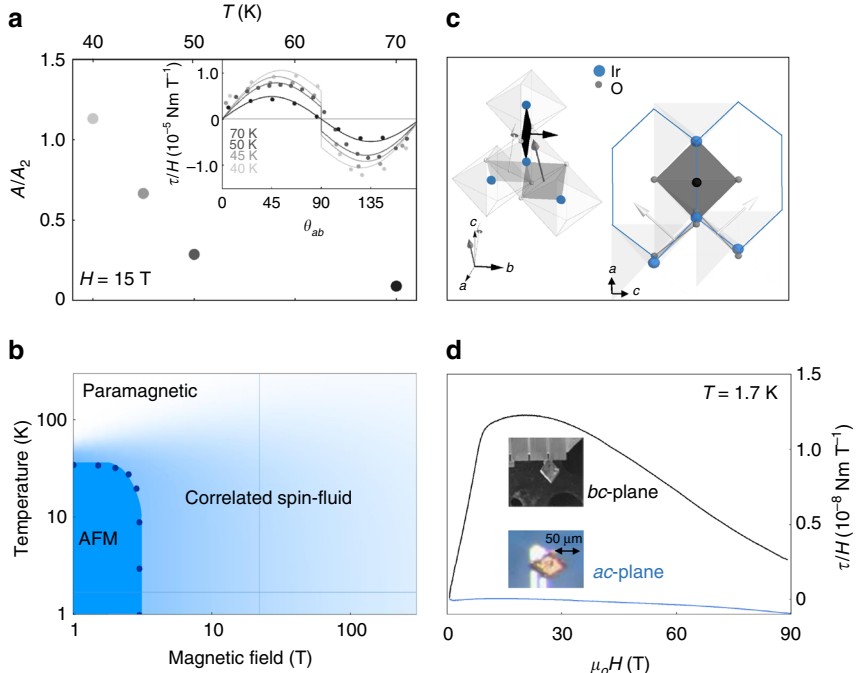

**Fig. 4** The temperature/magnetic field phase diagram and its relation to the spin-anisotropic exchange interactions of γ-lithium iridate. **a** The temperature evolution of the magnetic anisotropy represented by $A/A_2$. $A$ and $A_2$ are the coefficients determined by fitting the fixed-temperature angle dependence of the torque at fields above $H^*$ (inset) to $\tau/H = A \sin\theta \, \text{sign}(\cos\theta) + A_2 \sin 2\theta$. **b** Schematic representation of the temperature/magnetic field phase diagram of γ-lithium iridate. At low temperatures, the ordered phase (*dark blue*) extends to H = 3 T for magnetic fields along the b-axis. The correlated spin-fluid (*light blue*) likely persists to magnetic fields above 100 T. The faint lines represent regions of the phase diagram explored with constant field and temperature values in **a** and **d**, respectively. **c** The iridium-iridium exchange pathways mediated by oxygen octahedra. The principle components of the exchange tensor $J_{ij}$ between two iridium ions (defined via the exchange energy $J_{ij}S_iS_j$) are determined by the orientation the Ir-O$_2$-Ir plaquette which connects them together. The arrows represent the direction perpendicular to the Ir-O$_2$-Ir plaquette for each of the nearest neighbor bonds. The view along the b-axis illustrates that only one orientation of the Ir-O$_2$-Ir planes is perpendicular to a crystal direction. This is the only direction for which the exchange interaction is finite in the Kitaev analysis of the ground state of this complex spin system. **d** $\tau/H$ shown for two principal components of magnetic anisotropy ($\alpha_{bc}$ and $\alpha_{ac}$ up to 90 T, highlighting the special role of $M_b$

(Supplementary Information of ref. [4]). The sin 2θ dependence below $H^*$ is in stark contrast with the angle dependence at high fields, where torque exhibits a sharp discontinuity as the magnetic field crosses the c-axis (Fig. 3b). We observe the nonlinear susceptibility at fields above $H^*$ as a sin θ component, accompanied by a sign(cos θ) factor that captures the observed discontinuity at 90°. The sin θ angle dependence indicates that the torque $\tau = M_bH_c - M_cH_b$ in this high-field regime is dominated by the first term, where $H_c = H \sin\theta$. The negligible contribution of the second term $M_cH_b$ is confirmed by the observed torque in the hc-plane: if the discontinuity in $\tau/H$ vs. θ is driven exclusively by the saturated b-component of magnetization, then one would expect the amplitude of sin θ in the hc-plane to be reduced by a factor cos 55° ≈ 0.577 compared to the bc-plane. As expected, the reduced amplitude factor is 0.573. Thus, $M_b$ dominates the total magnetization at very high fields. Futhermore, the high-field magnetic response indicates that the Ising-like discontinuity $M_b = M_b^*\text{sign}(H_b)$ across the c-axis extends well beyond $H^*$.

In the ultra high-field limit, when all spins are nearly aligned with the applied magnetic field, the effective local anisotropy energy (per formula unit) is $E \approx (\mu_{\text{Ir}}^2\beta/2)\cos 2\theta$, where $\mu_{\text{Ir}}$ is the magnetic moment of the iridium ion and $\beta$ is the anisotropic spin stiffness. Therefore when all correlations are overcome by a very large applied magnetic field, the torque $\tau = dE/d\theta \approx \mu_{\text{Ir}}^2\beta\sin 2\theta$ is expected to recover the sin 2θ angle dependence observed at low fields. By contrast, in γ-lithium iridate, the sin θ sign(cos θ) angle dependence of the torque persists up to the highest applied magnetic fields (Fig. 3c), indicating the presence of robust spin

correlations throughout the entire field range and providing a lower bound for the magnitude of the exchange interactions.

To assess the extent of this correlated spin-fluid, we examine the thermal evolution of the nonlinear angle dependence of the torque at fields above $H^*$ (vertical line in Fig. 4b). Figure 4a shows the angle dependence of $\tau/H$ at 15 T for a range of temperatures, revealing a gradual decrease of the nonlinear response as temperature is raised. The nonlinearity becomes undetectable at this magnetic field above 70 K, the same temperature where paramagnetic behavior onsets in magnetic susceptibility measurements[4], indicating a continuous crossover to a strongly correlated spin-fluid at low temperatures. This crossover is similarly captured by a smoothly evolving torque signal upon cooling through the transition temperature at fields above $H^*$ (Fig. 1d). We note that specific heat measurements also observe a broad peak upon cooling at high fields[17], consistent with a decrease in entropy due to the onset of spin correlations[18]

## Discussion

The orientation of the spin-anisotropic exchange interactions with respect to the crystal directions gives rise to a strong magnetic anisotropy in the ordered state[4]. The extreme softening of $\chi_b$ occurs because the b-axis is the only direction coaligned either parallel or perpendicular to all Ir-O$_2$-Ir planes. None of the Ir-O$_2$-Ir planes are parallel or perpendicular to either the a- or c-axis (Fig. 4c). In this study, we find that the b-component of magnetization continues to dominate the magnetic response beyond the ordered state to the highest measured fields (Fig. 4d).

The high-field anisotropy bears resemblence to the broken symmetry state observed at low temperatures and low fields. This may suggest that the correlated object is a sub-unit of the complex magnetic structure observed at zero field: $H^*$ signifies the destruction of long-range order while leaving most of the local magnetic correlations intact. Perhaps the counter-propagating spin spirals seen by X-ray scattering in $\beta$-lithium iridate[6], are decoupled by a small component of magnetic field along the $b$-direction, leading to a finite correlation length with only minimal polarization of the individual spins.

The unusual behavior of the spin-fluid directly indicates that the observed magnetic anisotropy is driven by exchange interactions, rather than $g$-factor anisotropy that is tied to the honeycomb planes (Supplementary Information of ref. [4]). This observation, coupled with the anomalously small magnetic moment induced for all field orientations, identifies spin correlations that persist over a broad field and temperature range. In this context, we note that many conventional, as well as correlated, metals are unstable at low temperatures and undergo symmetry breaking that gaps their low-energy excitations[19, 20]. In frustrated magnets, long-range order can be stabilized by lattice distortions, crystal-field effects, and alternative exchange pathways. In the specific case of embedding the Kitaev model onto a 3D honeycomb lattice, it appears that long-range order is stabilized by intrinsic symmetry breaking—whereby only one component of the spin-anisotropic exchange can be coaligned with a crystal direction. Recent dynamic studies in related compounds[15, 16] have found exotic gapless excitations persisting after magnetic order is suppressed with relatively small magnetic fields. Although the ordered state at low temperatures in $\gamma$-lithium iridate has quelled the possibility of a Kitaev spin-liquid ground state, we have shown that this system hosts an exotic spin state that is otherwise masked by the zero-field antiferromagnetic order. Other studies, such as nuclear magnetic resonance[16], inelastic neutron scattering[15] or linear thermal transport in the zero-temperature limit[21] are necessary to determine whether the spin-fluid in $\gamma$-lithium iridate inherits any properties of a gapless spin-liquid.

**Data availability.** All relevant data is available from the authors.

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

## Acknowledgements
This work was performed at the National High Magnetic Field Laboratory, and was supported by the US Department of Energy BES 'Science at 100 T', the National Science Foundation DMR-1157490, and the State of Florida. R.M. acknowledges support from LANL LDRD DR20160085 'Topology and Strong Correlations'. J.A. and N.B. acknowledge support by the Department of Energy Early Career program, Office of Basic Energy Sciences under Contract No. DE-AC02-05CH11231. J.A. and N.B. also acknowledge support from the Gordon and Betty Moore Foundation's EPiQS Initiative through Grant GBMF4374.

## Author contributions
K.A.M., B.J.R., J.B.B., R.D.M. and A.S. performed the experiments at the National High Magnetic Field Laboratory-Pulsed Field Facility. K.A.M., B.J.R., R.D.M. and A.S. analyzed the data and wrote the manuscript with input from all authors. N.P.B. and J.G.A. synthesized and characterized high-quality single crystals of $\gamma$-lithium iridate at the University of California, Berkeley.

## Additional information

**Competing interests:** The authors declare no competing financial interests.

