## [Peer Review File · Nature Communications]

Reviewers' Comments:

Reviewer #1 (Remarks to the Author):

It is nice to read this manuscript and know the latest research work on harmonic honeycomb compound Li_2IrO_3 which author reported on Nature Communication 2014, 5, 4203. Such as Figure 1a which was recorded at 4K could be treated as an extended plot of Figure 4b of formal paper recorded at 1.5K because both are lower than magnetic phase transition temperature at 38K. I recommend this manuscript to be published after minor revision:

1. It is better to change the title from "...in the honeycomb lattice" to "... in the harmonic honeycomb lattice". Because it is not a two-dimensional layered honeycomb lattice, it is a three-dimensional harmonic honeycomb lattice.
2. On Page 4, line 5 from bottom: it is necessary to supply the related data of specific heat measurement (mentioned on reference 17) as a broad peak at high field to confirm the existence of the spin correlation.

Reviewer #2 (Remarks to the Author):

The manuscript consider the effects of (extremely) strong magnetic fields on a three-dimensional honeycomb material which is thought to have dominant "Kitaev" interactions. The authors find that such a field, with appropriate direction, can destroy the magnetic ordering that is present at zero field. They characterize the state that arises after this destruction through an analysis of the magnetic torque, looking in detail at its dependence on field direction. They find that this state is inconsistent with expectations from the infinite-field limit; that is the magnetic moment is far from saturated and the angle dependence of the torque does not agree with naive expectations. While they do not draw definite conclusions, they do suggest that this field induced state could be related to the Kitaev spin liquid that fails to appear at zero field or to complex incommensurate spirals that appear at low fields.

Overall my impression of this manuscript is somewhat positive. The results are intriguing and the interpretation is mostly conservative. The authors have shown clearly that there is a distinct intermediate regime between the low-field ordered state and the expected polarized state at high enough fields. Given the expected exchange scales in these systems, one may not have expected to reach such a "polarized" limit -- but the nature of this intermediate state itself is of interest.

The authors have shown that this intermediate state has some clear properties that are very different from either limits; namely the importance of the easy-axis field component. Further, the lack of features in the temperature dependence of the torque at 10T, or in the field dependence at low temperature, seems to suggest this state smoothly connects to both the paramagnetic regime and the polarized regime (which is not accessible in these experiments). The authors state that is consistent with specific heat measurements at 10T, where again only broad features are seen (this is only stated in the text, with a reference to an unpublished manuscript). Together, this would suggest that this state does not break any additional symmetries (other than those explicitly broken by the magnetic field). Given the low symmetry of the crystal, this may still allow for some freedom in the kind of state stabilized (e.g. some kind of complicated ferrimagnet).

However, I think this work could motivate further studies (both theoretical and experimental) of the high-field physics of these materials. The details of this field-induced phase may provide some insights into the zero-field physics and even possibly help understand how to better approach the Kitaev spin liquid itself. Given the handful of Kitaev materials, and that all of them exhibit magnetic ordering, this is the kind of new information (going to high enough fields to strongly leave the linear regime) that is needed to move this field forward.

I have a few hopefully minor comments for the authors which I list below. If they can be addressed then I think I could recommend publication of in Nature Communications.

Comments/concerns:

- This may be a naive question: in the measurement of the torque could the discontinuity for $H > H^*$ some kind of sample re-orientation in the field?

- In Fig 1a it would be useful to have some explicit labels for the different angles. As it stands one can not determine the angle for the different colors, save by cross-referencing with Fig 2a, which is not ideal.

- The authors use the term "correlated spin fluid" in the title and in several places in the text. In my opinion, it somewhat is unclear what points to this state being "correlated", rather than some complicated "uncorrelated" magnetic state or some kind of simple ferrimagnet. Or is this referring simply to the that the strong b-axis anisotropy is likely due to exchange coupling? Or perhaps this discontinuity in the torque? (i.e. not g-factors) I do not think the physics being exchange dominated is sufficient to refer to it this way (at these field one expects it to be exchange dominated)

Some clarification is needed on which experimental features are being interpreted as evidence

for a "correlated spin fluid" vs. (say) some kind of ferrimagnet.

- The authors state (page 5, final paragraph)

"A spin-liquid ground state is characterized by a lack of long-range order and gapless excitations"

This is not correct. Gapless excitations are not required; gapped spin liquids exist. The authors should correct this statement or should specialize it to the Kitaev spin liquid on this lattice, which is expected to be gapless (with sufficiently isotropic Kitaev couplings)

Reviewer #3 (Remarks to the Author):

In this communication, the authors present a torque magnetometry study of spin correlations and anisotropy in gamma-phase Li_2IrO_3 . This system undergoes a magnetic ordering transition near 38 K and has been previously explored due to its potential anisotropic Kitaev exchange terms in its underlying spin Hamiltonian. Many of the same authors have published a previous study exploring spin anisotropy in this system, but at lower magnetic fields. In this work, they push the field boundary significantly higher in order to explore the magnetic system's response.

The key takeaway from their torque measurements is that the spins in this material remain correlated beyond the critical field H^* , beyond which the non-coplanar zero field AF state is presumed to collapse. In particular, the authors find that the b component of the total magnetization dominates the response both in the ordered and in the field-polarized state. This is argued as proof that exchange interactions drive the anisotropy rather than single ion g-factor effects. The fact that anisotropic exchange continues to dominate over applied fields up to 60T is then one of the key experimental findings and a signature of a robustly correlated high field state. The second key finding is the inference of a correlated spin fluid in this high field regime.

I currently have two major reservations regarding the manuscript's conclusions, which currently preclude my recommendation for its publication in Nature Communications.

The first is that the significance of a dominant anisotropic exchange in this material isn't clear. Certainly in a Kitaev system, this is what you want. However the manuscript makes no mention of how other anisotropic exchange terms can be excluded? Pseudodipolar exchange terms in iridates and 5d transition metal oxides can far exceed the fields utilized here. How does one differentiate this more common effect observed in many existing iridates from a more exotic Kitaev term?

The second is that the manuscript really provides no justification for the use of the term “correlated spin fluid” in the high field state. What precludes this from being a glass or a short-range ordered AF state? If you are tuning the balance of exchange parameters, entering such a state is certainly possible and similar frozen states are commonly observed in frustrated materials. How can the authors demonstrate that the correlations are indeed liquid-like absent some probe of their dynamics?

Minor comments:

The statement “However, the particular bond-specific Ising exchange interactions considered in the Kitaev model have not been directly verified.” seems to ignore work probing anisotropic exchange in Na₂IrO₃ paper Ref. [4]. I believe that the authors mean a perfectly balanced Kitaev liquid hasn’t been realized, but perhaps that sentence could use some additional qualifiers.

The portion of the manuscript referencing unpublished specific heat data is not useful for the paper’s arguments. Absent actually being able to consider the specific heat data across the transition, referencing this in the discussion section is really a distraction for the reader.

To summarize, the manuscript does not currently meet the requirements for publication in Nature Communications, yet the claims made in the paper are of sufficient interest to the community if their novelty/importance can be properly justified.

Reviewer #4 (Remarks to the Author):

The authors of the paper present an experimental study, using torque magnetometry, of the magnetic response of gamma lithium iridate. This work extends an earlier experiment of Ref. [5]. The main findings of the present manuscript are the phase diagram Fig. 4b, and the collapse of $1/H^*$, shown in Fig. 2b.

The results presented in this paper should be of interest to people working in the areas of frustrated magnetism, and strongly-correlated electrons, and I suggest the paper for publication in Nature Communications.

Questions/comments

1. The nature of "correlated spin-fluid state" in the phase diagram Fig. 4b requires some discussion/clarification.
2. Does the collapse of $1/H^*$ dependence on H_b suggest that this component of the field

effectively decouples the system into 1D spin-chains running within the honeycomb planes, so that the "correlated spin-fluid state" is essentially one-dimensional?

3. Is there data on H^* dependence on temperature? This may be helpful in estimating the scale of non-Kitaev contributions to the effective magnetic Hamiltonian.

4. It would be useful to add a short supplementary material, perhaps reproducing a part of the discussion in Supp. Mat. of Ref. [5] related to the theory, or just to provide a reference to Supp. Mat. of Ref. [5]

5. The readability of Fig. 4a may be improved.

Reviewers' comments:

Reviewer #1 (Remarks to the Author):

It is nice to read this manuscript and know the latest research work on harmonic honeycomb compound Li_2IrO_3 which author reported on Nature Communication 2014, 5, 4203. Such as Figure 1a which was recorded at 4K could be treated as an extended plot of Figure 4b of formal paper recorded at 1.5K because both are lower than magnetic phase transition temperature at 38K. I recommend this manuscript to be published after minor revision:

1. It is better to change the title from "...in the honeycomb lattice" to "... in the harmonic honeycomb lattice". Because it is not a two-dimensional layered honeycomb lattice, it is a three-dimensional harmonic honeycomb lattice.

- **Thank you for your comments. We have changed the title to include "harmonic honeycomb iridates".**

2. On Page 4, line 5 from bottom: it is necessary to supply the related data of specific heat measurement (mentioned on reference 17) as a broad peak at high field to confirm the existence of the spin correlation.

- **The specific heat measurements are now properly referenced. To provide further support for spin correlations, we included reference to the measured torque upon cooling through the transition at high fields (Figure 1c). This dataset provides similar information as the specific heat data.**

Reviewer #2 (Remarks to the Author):

The manuscript consider the effects of (extremely) strong magnetic fields on a three-dimensional honeycomb material which is thought to have dominant "Kitaev" interactions. The authors find that such a field, with appropriate direction, can destroy the magnetic ordering that is present at zero field. They characterize the state that arises after this destruction through an analysis of the magnetic torque, looking in detail at its dependence on field direction. They find that this state is inconsistent with expectations from the infinite-field limit; that is the magnetic moment is far from saturated and the angle dependence of the torque does not agree with naive expectations. While they do not draw definite conclusions, they do suggest that this field induced state could be related to the Kitaev spin liquid that fails to appear at zero field or to complex incommensurate spirals that appear at low fields.

Overall my impression of this manuscript is somewhat positive. The results are intriguing and the interpretation is mostly conservative. The authors have shown clearly that there is a distinct intermediate regime between the low-field ordered state and the expected polarized state at high enough fields. Given the expected exchange scales in these systems, one may not have expected to reach such a "polarized" limit -- but the nature of this intermediate state itself is of interest.

The authors have shown that this intermediate state has some clear properties that are very different from either limits; namely the importance of the easy-axis field component. Further, the lack of features in the temperature dependence of the torque at 10T, or in the field dependence at low temperature, seems to suggest this state smoothly connects to both the paramagnetic regime and the polarized regime (which is not accessible in these experiments). The authors state that is consistent with specific heat measurements at 10T, where again only broad features are seen (this is only stated in the text, with a reference to an unpublished manuscript). Together, this would suggest that this state does not break any additional symmetries (other than those explicitly broken by the magnetic field). Given the low symmetry of the crystal, this may still allow for some freedom in the kind of state

stabilized (e.g. some kind of complicated ferrimagnet).

However, I think this work could motivate further studies (both theoretical and experimental) of the high-field physics of these materials. The details of this field-induced phase may provide some insights into the zero-field physics and even possibly help understand how to better approach the Kitaev spin liquid itself. Given the handful of Kitaev materials, and that all of them exhibit magnetic ordering, this is the kind of new information (going to high enough fields to strongly leave the linear regime) that is needed to move this field forward.

I have a few hopefully minor comments for the authors which I list below. If they can be addressed then I think I could recommend publication of in Nature Communications.

Comments/concerns:

- This may be a naive question: in the measurement of the torque could the discontinuity for $H > H^*$ some kind of sample re-orientation in the field?

- **Thank you for your comments. The observed discontinuity in the angle dependence at fields above H^* is due to a reorientation of the lever in magnetic field. This arises because the sample tries to avoid magnetic fields aligned along the c-axis (the magnetically hard direction at low temperatures). The sample is attached to the lever with silicon grease, frozen into place at low temperatures. We have never observed a reorientation of the crystal on the lever after the experiment. The angle dependence of the linear behavior at low fields is a good indication that we are applying field at the correct angle (w.r.t. the crystal axes) and we have no reason to believe the sample could suddenly begin to move on the lever at higher fields. Such behavior would probably result in a constant torque as a function field, which we do not observe.**

- In Fig 1a it would be useful to have some explicit labels for the different angles. As it stands one can not determine the angle for the different colors, save by cross-referencing with Fig 2a, which is not ideal.

- **A color bar and legend has been added to Figure 1a to directly reflect the angles for each field sweep, along with an updated caption.**

- The authors use the term "correlated spin fluid" in the title and in several places in the text. In my opinion, it somewhat is unclear what points to this state being "correlated", rather than some complicated "uncorrelated" magnetic state or some kind of simple ferrimagnet. Or is this referring simply to the that the strong b-axis anisotropy is likely due to exchange coupling? Or perhaps this discontinuity in the torque? (i.e. not g-factors) I do not think the physics being exchange dominated is sufficient to refer to it this way (at these field one expects it to be exchange dominated)

- **Your questions here acutely identify and summarize the complexities of the observed high-field state. Our use of the term "correlated spin-fluid" was our best attempt to capture these new complexities in a single term, while at the same time emphasizing a clear distinction from—and possible connection to—the spin-liquid state. Our data provides evidence that long-range order does not exist at high fields. Furthermore, since the magnetization is much smaller than the full $\sim 1\mu\text{B}$ moment at all fields, there is evidence that some kind of *locally* correlated magnetic structure with a finite correlation length is responsible for the observed behaviour. Because this object can be easily influenced by magnetic fields along one crystal direction, and shows no signs of hysteresis, we believe it also cannot be accurately described as glassy. Following your comment, we have clarified our use of this terminology in the text.**

Some clarification is needed on which experimental features are being interpreted as evidence for a "correlated spin fluid" vs. (say) some kind of ferrimagnet.

- **“Spin -fluid” refers to the correlated spin state that is not glassy (we observe no hysteresis and the magnetic anisotropy strongly resembles that of the ordered state) and that lacks broken symmetry (we observe a compensated net moment even at the highest fields whose peculiar Ising-like anisotropy is tied to the lattice directions — and therefore mediated by exchange interactions—and we observe no further transitions even up to 90 Tesla). With this term, we wanted to leave open the exciting possibility that the correlated state exposed with magnetic field is spin-liquid-like. Such suggested behavior is now further supported by evidence for fractionalized excitations in the state revealed with magnetic field in alpha-RuCl3 (references 15 and 16 in the updated version), which shares many similarities to the honeycomb iridates.**

- The authors state (page 5, final paragraph)

"A spin-liquid ground state is characterized by a lack of long-range order and gapless excitations"

This is not correct. Gapless excitations are not required; gapped spin liquids exist. The authors should correct this statement or should specialize it to the Kitaev spin liquid on this lattice, which is expected to be gapless (with sufficiently isotropic Kitaev couplings)

- **Thank you for your careful reading. We have updated this in the text.**

Reviewer #3 (Remarks to the Author):

In this communication, the authors present a torque magnetometry study of spin correlations and anisotropy in gamma-phase Li₂IrO₃. This system undergoes a magnetic ordering transition near 38 K and has been previously explored due to its potential anisotropic Kitaev exchange terms in its underlying spin Hamiltonian. Many of the same authors have published a previous study exploring spin anisotropy in this system, but at lower magnetic fields. In this work, they push the field boundary significantly higher in order to explore the magnetic system's response.

The key takeaway from their torque measurements is that the spins in this material remain correlated beyond the critical field H^* , beyond which the non-coplanar zero field AF state is presumed to collapse. In particular, the authors find that the b component of the total magnetization dominates the response both in the ordered and in the field-polarized state. This is argued as proof that exchange interactions drive the anisotropy rather than single ion g-factor effects. The fact that anisotropic exchange continues to dominate over applied fields up to 60T is then one of the key experimental findings and a signature of a robustly correlated high field state. The second key finding is the inference of a correlated spin fluid in this high field regime.

I currently have two major reservations regarding the manuscript's conclusions, which currently preclude my recommendation for its publication in Nature Communications.

The first is that the significance of a dominant anisotropic exchange in this material isn't clear. Certainly in a Kitaev system, this is what you want. However the manuscript makes no mention of how other anisotropic exchange terms can be excluded? Pseudodipolar exchange terms in iridates and 5d

transition metal oxides can far exceed the fields utilized here. How does one differentiate this more common effect observed in many existing iridates from a more exotic Kitaev term?

- **Thank you for your comments.** In our previous study (ref 5), the temperature dependence of the magnetic anisotropy provides evidence for anisotropic exchange interactions between the spins. At high temperatures, the observed magnetic anisotropy is captured by considering only g-factor anisotropy and the relative orientation of the two honeycomb planes. At low temperatures, we observe a ten-fold increase in the magnetic susceptibility along the b-axis, which can only be explained by the fact that only one of the three nearest neighbor Ir-Ir bonds has a Kitaev component of the exchange interaction that coaligns with the b-direction and there are more of these types of bonds than the other two orientations of Ir-Ir bonds in the lattice. The octahedral distortion is negligible in χ -Li₂IrO₃, which means that the exchange mechanism via intervening oxygen is approximately the same for all nearest neighbor iridium spins. Such a scenario, combined with the lack of a structural phase transition at all temperatures, makes it difficult to explain the large increase in χ_b at low temperatures without considering spin-anisotropy in the exchange interactions themselves. By a similar argument, pseudodipolar exchange terms would result in the same spatial anisotropy for all nearest neighbor links due to the locally identical environment between all Ir-Ir bonds. This would not give rise to one special direction, but would instead look like the magnetic anisotropy tied to the orientation of the honeycomb planes observed at high temperatures. We want to emphasize that the important point in this paper is the fact that the b-component of magnetization continues to dominate the anisotropic response well beyond the suppression of long-range order, which is interesting in its own right. Unfortunately, our data does not provide us with any microscopic details regarding the origin of this behavior, which may be useful in determining the relevant anisotropic exchange terms.

The second is that the manuscript really provides no justification for the use of the term “correlated spin fluid” in the high field state. What precludes this from being a glass or a short-range ordered AF state? If you are tuning the balance of exchange parameters, entering such a state is certainly possible and similar frozen states are commonly observed in frustrated materials. How can the authors demonstrate that the correlations are indeed liquid-like absent some probe of their dynamics?

- **We appreciate your questions here, which accurately identify and summarize the complexities of the observed high-field state.** Our use of the term “correlated spin-fluid” was our best attempt to capture these new complexities in a single term, while at the same time emphasizing a clear distinction from—and possible connection to—the spin-liquid state. This suggestion may be further supported by evidence for fractionalized excitations in the state revealed with magnetic field in alpha-RuCl₃ (references 15 and 16 in the updated version), which shares many similarities to the honeycomb iridates. Following your comments, we have clarified use of this terminology in the text.

“Spin -fluid” refers to the correlated spin state that is not glassy (we observe no hysteresis and the magnetic anisotropy strongly resembles that of the ordered state) and that lacks broken symmetry (we observe a compensated net moment even at the highest fields whose peculiar ising-like anisotropy is tied to the lattice directions — and therefore mediated by exchange interactions—and we observe no further transitions even up to 90 Tesla). Because this correlated object can be easily influenced by magnetic fields along one crystal direction, and shows no signs of hysteresis, we believe it also cannot be accurately described as glassy. Our evidence for no long-range order comes from the lack of broken symmetry in the high-field state. A new broken symmetry phase would have a well-defined boundary in the B-T phase diagram. We do not detect a sharp boundary in the broad range of temperatures and magnetic fields in this study. In the absence of long-range order, the correlations are “liquid-like”. Although we do not have evidence for the peculiar dynamics in this state, we do have evidence that (i) the correlated state at high fields is controlled by exchange interactions (rather than g-factor anisotropy) and (ii) the exchange interactions are strongly spin-anisotropic because only one component (b — the only one co-aligned with crystallographic directions) exhibits a significant magnetization response.

Minor comments:

The statement "However, the particular bond-specific Ising exchange interactions considered in the Kitaev model have not been directly verified." seems to ignore work probing anisotropic exchange in Na₂IrO₃ paper Ref. [4]. I believe that the authors mean a perfectly balanced Kitaev liquid hasn't been realized, but perhaps that sentence could use some additional qualifiers.

- **We agree that the sentence "However, the particular bond-specific Ising exchange interactions considered in the Kitaev model have not been directly verified" was in disagreement with the sentence directly preceding it. This now reads "Kitaev's spin-liquid ground state, however, results from the presence of exclusively bond-specific Ising exchange interactions, which have not yet been verified experimentally."**

The portion of the manuscript referencing unpublished specific heat data is not useful for the paper's arguments. Absent actually being able to consider the specific heat data across the transition, referencing this in the discussion section is really a distraction for the reader.

- **The specific heat data is now properly referenced. In the discussion, we now refer to the measured torque upon cooling through the transition at high fields (Figure 1c), in support of spin correlations that emerge directly from the high temperature phase when magnetic fields are greater than H^* .**

To summarize, the manuscript does not currently meet the requirements for publication in Nature Communications, yet the claims made in the paper are of sufficient interest to the community if their novelty/importance can be properly justified.

Reviewer #4 (Remarks to the Author):

The authors of the paper present an experimental study, using torque magnetometry, of the magnetic response of gamma lithium iridate. This work extends an earlier experiment of Ref. [5]. The main findings of the present manuscript are the phase diagram Fig. 4b, and the collapse of $1/H^*$, shown in Fig. 2b.

The results presented in this paper should be of interest to people working in the areas of frustrated magnetism, and strongly-correlated electrons, and I suggest the paper for publication in Nature Communications.

Questions/comments

1. The nature of "correlated spin-fluid state" in the phase diagram Fig. 4b requires some discussion/clarification.

- **Thank you for your comments. We agree that our use of the term "correlated spin-fluid" was not properly justified. Following your comment, we have clarified our use of this terminology in the main text.**

2. Does the collapse of $1/H^*$ dependence on H_b suggest that this component of the field effectively

decouples the system into 1D spin-chains running within the honeycomb planes, so that the "correlated spin-fluid state" is essentially one-dimensional?

- **Indeed, we wanted to provide one possible scenario that could be the reason for the observed behavior. In this picture, small magnetic fields along the b-direction decouple spins between the zigzag chains, while spins within a chain are tightly locked and give rise to the observed magnetic anisotropy. However, our data cannot rule out other possibilities (ie. spins may be more correlated between the zigzag chains rather than in them).**

3. Is there data on H^* dependence on temperature? This may be helpful in estimating the scale of non-Kitaev contributions to the effective magnetic Hamiltonian.

- **Yes - below the magnetic ordering temperature, H^* depends very little on temperature. This is can be seen in Figure 1c on cooling through the transition at various fields.**

4. It would be useful to add a short supplementary material, perhaps reproducing a part of the discussion in Supp. Mat. of Ref. [5] related to the theory, or just to provide a reference to Supp. Mat. of Ref. [5]

- **We agree that the supplementary of reference 5 is a helpful tool for the reader of this manuscript. It is particularly useful for the discussion that allows us to rule out the significance of g-factor anisotropy at high fields. We have included reference to the supplementary information of reference 5 in this part of the discussion, as well as other places throughout the text.**

5. The readability of Fig. 4a may be improved.

- **We have reworked Figure 4 and we hope this improves readability.**

Reviewers' Comments:

Reviewer #1 (Remarks to the Author):

I am satisfied of authors's reply to all of comments from reviewers. I recommend it to be published on Nature Communication.

Reviewer #2 (Remarks to the Author):

In their response the authors have addressed most of the issues raised in my report in a satisfactory manner and have clarified some of the issues I had with the work in the main text. I have no further reservations and thus recommend publication in Nature Communications.

One small comment: I think, for the sake of reader, it would be useful to mention the origin of the angular discontinuity mentioned in the authors' response [1] somewhere in the manuscript itself.

[1]: "The observed discontinuity in the angle dependence at fields above H^* is due to a reorientation of the lever in magnetic field"

Reviewer #3 (Remarks to the Author):

The revised manuscript submitted by the authors addressed my initial concerns regarding the manuscript's discussion of the key experimental results. The amended presentation also makes a clearer case for an unusual spin state that persists at high fields in gamma-phase Li_2IrO_3 . Although the use of the term "fluid" to refer to the spins in this material remains poorly justified in light of the myriad of unusual frozen states known to emerge in frustrated magnets, I do recognize that the term is often used in similar contexts. My parting recommendation for the authors is to consider an alternative description for the high field state that more directly reflects what the data reveals.

I believe that the manuscript is now suitable for publication in Nature Communications.

Reviewer #4 (Remarks to the Author):

I am happy with the authors responses, and recommend the paper for publication in Nature Communications.

REVIEWERS' COMMENTS:

Reviewer #1 (Remarks to the Author):

I am satisfied of authors's reply to all of comments from reviewers. I recommend it to be published on **Nature** Communication.

Reviewer #2 (Remarks to the Author):

In their response the authors have addressed most of the issues raised in my report in a satisfactory manner and have clarified some of the issues I had with the work in the main text. I have no further reservations and thus recommend publication in **Nature Communications**.

One small comment: I think, for the sake of reader, it would be useful to mention the origin of the angular discontinuity mentioned in the authors' response [1] somewhere in the manuscript itself.

[1]: "The observed discontinuity in the angle dependence at fields above H^* is due to a reorientation of the lever in magnetic field"

This statement cannot be incorporated into the text as is because it is not technically correct. The discontinuity is ultimately due to the Ising-like response of the magnetization, which causes the lever to flop to either side of the crystal's magnetically hard-axis. The suggested rewrite implies that the observed behavior is a manifestation of the lever's response and not the sample's, which is not the case. The current version of the manuscript does a better job of describing the correct behavior.

Reviewer #3 (Remarks to the Author):

The revised manuscript submitted by the authors addressed my initial concerns regarding the manuscript's discussion of the key experimental results. The amended presentation also makes a clearer case for an unusual spin state that persists at high fields in gamma-phase Li_2IrO_3 . Although the use of the term "fluid" to refer to the spins in this material remains poorly justified in light of the myriad of unusual frozen states known to emerge in frustrated magnets, I do recognize that the term is often used in similar contexts. My parting recommendation for the authors is to consider an alternative description for the high field state that more directly reflects what the data reveals.

I believe that the manuscript is now suitable for publication in **Nature Communications**.

The authors have considered a number of alternative descriptions to spin-fluid, none of which we believe does an adequate job of capturing all of the experimental features. We would like to continue using spin-fluid to describe the high-field state because 1) this most closely describes our dataset and 2) it implies a possible connection between our work and very exciting recent studies that find evidence for a spin-liquid in related compounds.

Reviewer #4 (Remarks to the Author):

I am happy with the authors responses, and recommend the paper for publication in **Nature Communications**.